# Genome-Wide Analysis Reveals Human-Mediated Introgression from Western Pigs to Indigenous Chinese Breeds

**DOI:** 10.3390/genes11030275

**Published:** 2020-03-04

**Authors:** Jue Wang, Chengkun Liu, Jie Chen, Ying Bai, Kejun Wang, Yubei Wang, Meiying Fang

**Affiliations:** 1National Engineering Laboratory for Animal Breeding, MOA Laboratory of Animal Genetics and Breeding, Beijing key Laboratory for Animal Genetic Improvement, College of Animal Science and Technology, Department of Animal Genetics and Breeding, China Agricultural University, Beijing 100193, China; tctcttc@hotmail.com (J.W.);; 2Berry Genomics Corporation, Beijing 102206, China; singfw@cau.edu.cn; 3College of Life Sciences and Food Engineering, Hebei University of Engineering, Handan 056038, China; baiyinghelen@gmail.com; 4College of Animal Science and Veterinary Medicine, Henan Agricultural University, Zhengzhou 450002, China; wangkejun.me@163.com

**Keywords:** pigs, genome-wide variation, introgression, artificial selection

## Abstract

Genetic variations introduced via introgression from Western to Chinese pigs have contributed to the performance of Chinese breeds in traits such as growth rate and feed conversion efficiency. However, little is known about the underlying genomic changes that occurred during introgression and the types of traits affected by introgression. To address these questions, 525 animals were characterized using an SNP array to detect genomic regions that had been introgressed from European to indigenous Chinese breeds. The functions of genes located in introgressed regions were also investigated. Our data show that five out of six indigenous Chinese breeds show evidence of introgression from Western pigs, and eight introgressed genome regions are shared by five of the Chinese breeds. A region located on chr13: 12.8–13.1 M was affected by both introgression and artificial selection, and this region contains the glucose absorption related gene, *OXSM*, and the sensory related gene, *NGLY*. The results provide a foundation for understanding introgression from Western to indigenous Chinese pigs.

## 1. Introduction

Domesticated pigs are one of the most numerous large mammals on the planet [1,2] and have a wide range of distinct morphological and behavioral characteristics, as compared to their wild progenitors [3]. Based on current evidence, pigs were domesticated independently in Anatolia [4,5] and the Mekong Valley [6] about 9000 years before the common era (BCE). Thus, two domestication centers, located in Europe and Asia, have contributed to the modern pig germline.

Introgression is common in animals and has been reported widely [7,8,9,10,11,12]. It can happen not only between wild ancestors and their domesticated offspring, but also among domesticated animals. Introgression between wild and domesticated populations usually occurs in adjacent areas, as has been the case for pigs [8,13,14], sheep [9], dogs [11], and chickens [7,10,12]. Most introgression among domesticated animals has been the result of their strict management by humans [15]. For example, human-mediated introgression has been detected in cattle breeds [16].

Management effects, especially the crossbreeding between Asian and European pigs, have played an important role in pig breeding. Pig admixture between Asia and Europe became common in the mid-to-late 18th century [3]. Chinese Meishan pigs were introduced into Europe to improve reproductive traits [17], resulting in the development of modern breeds such as Yorkshire (i.e., Large White), Berkshire, and Hampshire [18,19]. In the late 18th century, Chinese pigs were also imported to America [14] and crossed with local pigs for performance improvement [20]. Variations in mitochondrial DNA have been used to confirm that indigenous Chinese pigs were imported into Europe and contributed to the development of European commercial breeds. The average introgression level was determined to be approximately 33% [21]. A few genome-wide studies have shown that introgression has also occurred from European to Asian pigs [22,23,24]. However, the genomic differences caused by introgression were not examined, and the functions affected by the altered genomic regions are also unknown.

In this study, 525 pigs from 20 breeds were characterized using an SNP array. The animals included individuals from six areas of China in which indigenous breeds are found. The data were used to analyze introgression from European to different Chinese breeds, and the introgressed regions were examined to identify genes and their associated functions. Introgressed genomic regions affected by artificial selection were also investigated to elucidate the ways in which European pigs contributed to the improvement of traits in indigenous Chinese breeds.

## 2. Materials and Methods 

### 2.1. Sample Collection and Quality Control

All samples were collected according to the Guidelines for the Protection and Use of Laboratory Animals established by the Chinese Ministry of Agriculture. DNA was extracted from ear tissue samples using a QIAGEN kit, following protocols provided by the manufacturer. Purified DNA was diluted to 20 ng/μL for genotyping [25]. A total of 278 samples from 13 breeds were genotyped using the Porcine SNP60 BeadChip (Illumina, San Diego, CA, USA), which detects over 64,000 SNPs [25]. The 278 samples represented 234 individuals from China (196 domesticated pigs and 38 wild boars from most parts of China; breeds and sources are shown in Table 1), 12 European wild boars, and 32 Duroc pigs (the SNP genotypes in VCF format were deposited in the figshare data repository: doi:10.6084/m9.figshare.11911377). In addition, 208 Western pigs and 39 warthogs (the warthogs were used as an outgroup) were obtained from an online database [26,27]. The geographical distribution of the samples is shown in Figure 1, and the samples are described in detail in Table 1. SNP characteristics for all samples were evaluated using PLINK v1.9 [28]. SNPs that either failed to pass the Minor Allele Frequency (MAF) test (1%) or were missing in over 5% of the SNPs in a sample were removed. After low-quality SNPs were excluded, 42,819 autosomal SNPs remained, representing 525 pigs and 20 breeds. Table 1 includes the number of samples collected from each breed/population, along with the breed abbreviations used throughout this report.

### 2.2. Population Structure

Principal component analysis (PCA) was conducted using GCTA software [29]. A scatterplot was generated to visualize the first and second principal components, based on a variance-standardized relationship matrix from the PCA results. Population assignment analysis was performed using ADMIXTURE [30] with the number of clusters (K) varying from 2 to 9 (10,000 iterations). The results from the analysis were interpreted using methods described by Evanno et al. [31].

Evolutionary distances were computed using the two-parameter method described by Kimura [32] and are expressed as the number of base substitutions per site. All positions containing gaps and missing data were eliminated. Evolutionary analyses were conducted using MEGAX [33].

### 2.3. Whole-Genome Analysis of Genomic Introgression

Patterson’s D-statistic [34,35] was used in our study to detect introgression from European pigs to indigenous Asian pigs. In a typical tree topology (((P1, P2), P3), O), where “O” refers to an outgroup, the null hypothesis is that P1 and P2 share the same ancestor with P3, and that there was no gene flow between P3 and either P2 or P1 after their ancestors diverged. If the results deviate from the null hypothesis, a positive D-value is expected when P2 shares more derived alleles with P3, while a negative D-value is expected when P1 and P3 have more alleles in common. Therefore, the D-statistic calculates the level and direction of introgression.

In our study, we calculated the D-statistic using different combinations of indigenous Asian pigs and European pigs. The D-statistic was established using (((ASW, AD), EU), warthog), in which warthog (*Phacochoerus*) served as the outgroup, ASW symbolizes Asian wild boars, AD is several indigenous Asian breeds including TIB (Tibetan pig), EHL (Erhualian pig), RC (Rongchang pig), WZS (Wuzhishan pig), JH (Jinhua pig), and MIN (Min pig) in 5 independent tests, and EU includes COM (commercial breeds, including LAD (Landrace), LW (Large White), and PIT (Pietrain)), EUD (European domesticated pigs, excluding commercial breeds), and DUR (Duroc) in different combinations.

To quantify the size of the window affected by introgression, the $f_{d}$ statistic [36] was calculated. Sliding windows contained 10 consecutive SNPs and were positioned at 2 SNPs intervals across every chromosome in the genome. The $f_{d}$ statistic was calculated using Dsuite. The *P*-value was evaluated from the Z-transformed $f_{d}$ value [37], and regions with *p*-value < 0.05 were classified as significantly introgressed genomic regions [38]. The formula for the Z-transform is Z=fD−μσ, where $f_{d}$ is the modified $f_{d}$-statistic, μ is expected value of $f_{d}$ for five introgressed populations, and σ is the standard deviation.

### 2.4. Selection Scan

We used XP-CLR [39] to scan the genome for selective signatures. XP-CLR (cross-population composite likelihood ratio) is a multilocus sliding window test that jointly models the multilocus allele frequency differentiation between two populations. The statistic is particularly robustly resistant to ascertainment bias and well-suited for population demography. This method was used to calculate whether introgressed regions were affected by artificial selection. Genetic distance was estimated by physical position, where 1 cM = 1 Mbp [40]. The parameters used for XP-CLR were -w1 1 20 200000 ${i} -p0 0. The sliding window size was 1 cM, with a 200-kb step size. A weighted CLR scheme was used to estimate XP-CLR. All XP-CLR scores were output to files. File format conversion was accomplished using the Ruby script “vcf2geno.rb”. Regions were ranked by score, and regions scoring in the top 5% in each comparison were classified as significantly selected.

Haplotypes networks were constructed in PopART version 1.7 [41] using default parameters to produce median-joining haplotype networks [42]. The haplotype number was counted by DnaSP6 [43]; the interprocess was done with R scripts [44].

## 3. Results

### 3.1. Inference of Population Structure and Degree of Breed Admixture

To characterize the relationships between the genetic backgrounds of Western and Chinese pigs, we first performed a population structure analysis. Warthogs were excluded from the analysis because large differences between this species and other populations might mask more subtle differences between Western and Chinese pigs. An analysis that includes warthogs is provided as a Appendix A), and warthogs were included as an outgroup in the construction of the phylogenetic tree (see below).

PCA results are shown in Figure 2a. Principal component 1 captures 14.30% of the variance, while principal component 2 captures 4.91%. Component 2 separates Duroc (c4) from other European breeds. European wild boars (c3) and European domestic pigs (c2) also formed their own clusters on this axis. In contrast, wild boars and domestic pigs from China (c1) are not separated on the second axis, but MIN and JH are unusual because they fall in the middle of the first principal component between two clusters. If an analysis is conducted using Asian breeds alone (Appendix A), wild boars and domestic pigs form distinct clusters. A neighbor-joining phylogenetic tree shows that different subgroup form clear clusters (Figure 2b).

The genetic structure of the population is shown in Figure 2c. When K (the assumed number of ancestors) is small (K = 2–4), Asian pigs, European domesticated pigs, European boars, and Duroc pigs are resolved. At K = 2, EUD and DUR are found together (in the red section), reflecting both their shared ancestry with Asian pigs and the fact that genetic resources from Asian breeds are included in commercial breeds, consistent with the PCA results. With increasing values of K (> 4), more details emerge in Asian pigs. The primary changes in clustering occur in Asian pigs. EHL and WZS cluster separately when K = 5 and K = 8, and the Asian wild boar clusters at K = 9. MIN and JH appear in a mixture with multiple ancestral sources.

### 3.2. Introgression from European Pigs to Chinese Indigenous Breeds

To obtain more detailed genomic evidence for introgression from European pigs to indigenous Chinese pigs, we calculated the D-statistics for each combination of Western and Chinese breeds. Western pigs were grouped as COM (European commercial breeds, including LAD, LW, and PIT), EUD (European domesticated pigs, excluding commercial breeds), DUR (Duroc), and EW (European wild boars). The results show that all of the indigenous Chinese pigs exhibited introgression, with the exception of RC (D = 1.9 × 10^−3^, *p*-value = 0.43). Because the D-statistics were positive (Table 2), the direction of introgression is from the Western pigs into the Chinese breeds. The large range in D-statistics indicates that different levels of introgression were detected among the Chinese pig breeds. To exclude the effects of gene flow from Asian domesticated pigs to European pigs (as mentioned before), we calculated the introgression from EW to AD. In the results, we found that five Chinese indigenous breeds were introgressed (MIN, JH, TIB, WZS, and EHL), which was consistent with our previous results. The most significant introgressions were observed from COM to MIN ((((ASW, MIN), COM), warthog), D = 0.45, *p*-value < 1.0 × 10^−26^) and from DUR to JH ((((ASW, JH), DUR), warthog), D = 0.29, *p*-value < 1.0 × 10^−26^).

To locate the introgressed genomic regions in the genomes of the indigenous Chinese pigs, we computed the modified f-statistic ($f_{d}$) value in a sliding window analysis, using windows containing 10 SNPs, a step size of 2 SNPs, and a cutoff of *p* < 0.05 after application of the Z-transform. The introgressed regions include 15.47% of the MIN genome, 7.63% of JH, 2.55% of TIB, 1.29% of WZS, and 1.08% of EHL. We then combined regions that were significantly affected by introgression in all five breeds, yielding eight merged regions distributed on six chromosomes (chr6: 22591224–24174352, chr7: 24451815–27317882, chr8: 4659298–5191196, chr12: 24607160–25035796, chr13: 11949152–13065904, chr14: 47696012–48655418, 101124119–103123266, and 109827501–111397655). Putative genes within the merged regions were identified using the genomic locations of SNP to recover information from the annotated pig genome. Among the 34 genes found were several that appear to be related to muscle growth (*REG3G, TNXB*), bone hyperplasia (*IER3, SGMS2, DKK2*), sensory organs (sight: *CFB, NGLY*, and hearing: *DDR1, USP53*), and digestive secretion (*PRDM5, NKX2-3, OXSM, TNF*).

### 3.3. Investigation of Introgressed Regions Affected by Artificial Selection

To identify genes possibly under artificial selection in Chinese pigs, we calculated the XP-CLR value between six domesticated indigenous Chinese breeds and Chinese wild boars using a 1 Mb sliding window with 200 kb steps respectively. With these parameters, the average window included 13.68 SNPs. Selective signals were required to fall within the top 5% of XP-CLR values. A total of 1220 selective signals were considered in each comparison. All six indigenous Chinese breeds exhibited significant levels of artificial selection in 11 genomic regions. If the analysis is constrained to the 5 introgressed breeds, the number of regions showing evidence of selection increases to 19. Manhattan plots for XP-CLR values representing all comparisons are shown in Figure 3a. Interestingly, the region spanning chr13: 12.5–13.1M has been subjected both to artificial selection and introgression in five of the six indigenous Chinese breeds, but RC shows no evidence for artificial selection. We hypothesize that this region was affected by introgression and was retained to improve performance after artificial selection in indigenous Chinese pigs. Two putative genes were identified in the region and have functions related to glucose absorption (*OXSM*) and sensory organs (*NGLY*).

We constructed a haplotype network [41] using the 15 haplotypes in chr13: 12.5–13.1Mb (Figure 3b). The ASW and RC haplotypes were identical in this region. Because RC has not been introgressed from Western commercial pigs, we believe that haplotype I is the original haplotype in Chinese breeds. The introgressed indigenous Chinese breeds are mainly represented by haplotype II, and we also found this haplotype in COM and ED. In addition, compared with Western pigs, the indigenous Chinese breeds have only a few haplotypes concentrated in this region (I, II, III, and IV), suggesting that the region has been subjected to artificial selection.

The population structure of MIN appears to be unusual in comparison with the other indigenous Chinese breeds. In particular, MIN had higher $f_{d}$ values than other breeds, indicating that it has been subjected to higher levels of introgression [24]. In order to investigate the unusual genomic characteristics of MIN in more detail, we calculated XP-CLR values between MIN and ASD (including the other indigenous Chinese local breeds EHL, JH, RC, TIB, and WZS) to identify MIN-specific regions and also calculated XP-CLR between MIN and COM, because of the markedly high D-statistics from COM to MIN. Examining the ratio of the mean XP-CLR in the two comparisons, the difference between MIN and COM is smaller than the difference between MIN and ASD. This indicates that MIN is more closely related to COM in the introgressed regions. However, some regions are significantly different between MIN and ASD but are not different between MIN and COM, implying that MIN had been severely introgressed. 

## 4. Discussion

Pigs were domesticated in Europe and Asia independently, and research has detected gene flows between European and Asian pigs [8]. Our results confirmed that introgression has occurred from Western pigs to indigenous Chinese breeds. Moreover, the introgression levels vary among breeds, perhaps as a consequence of recent Chinese strategies and policies for regional introduction of germplasm. Chinese historical records show that Western pigs, most often Large Yorkshire, Berkshire, and Duroc, began to be introduced around 1840 [45]. In 1910, “white pigs” from the Russian Empire were introduced to the northeastern region of China for crossbreeding with local pigs [46]. In 1919, Lingnan University introduced Berkshire pigs to learn about the characteristics of European breeds and began to improve the performance of indigenous breeds through crossbreeding [45]. We suggest that these and later introductions into specific areas are responsible for the varied levels of introgression that we observe from Western to indigenous Chinese breeds.

It is particularly interesting that the Chinese Min breed has been subjected to a high level of introgression. This is likely to be a consequence of location, since these pigs inhabit the northeast border areas of China, adjacent to the Russian border. According to breeding records, a large number of commercial pigs were bred in Russia and imported into China [47], which is consistent with our finding that MIN has received an introgression from commercial pigs. Considering the high proportion of introgressed genes in MIN (14.57%), we speculate that MIN has been affected by more genetic change than can be attributed to introgression alone, such as crossbreeding. The level of introgression is higher than observed for some species. For example, Neanderthal introgression affected all non-African populations and is estimated at 1.8–2.4% in European and 2.3–2.6% in East Asian populations [35]; introgression from polar bears to North American brown bears is 3–8% [48,49]. However, the introgression in MIN is lower than from dogs to the Eurasian wolf (25%) [50,51].

Among the indigenous Chinese breeds, JH was the second-most affected by introgression. JH pigs are also located in China’s border areas, but they may have been affected by introgression from pigs imported via ocean routes [45]. The only indigenous Chinese breed that has not been affected by introgression is RC, found in the Sichuan Province in the inland area of China. We infer that pigs from border areas are more likely to be introgressed than pigs from internal regions.

Although the SNP array was not designed to identify specific genes, our analysis of introgressed regions revealed genes related to “skeletal development”, “muscle growth”, “digestive secretion”, and “senses”. These are associated with several important traits. For example, *REG3G, IER3, SGMS2,* and *DKK2* [52,53,54,55] are involved in skeletal development, *TNXB* [56] in muscle growth, *CH25H, DRG1* [57,58] in digestive secretion, and *NGLY* [59] in vision. The genes *OXSM* and *NGLY* are also located in regions affected by both introgression and artificial selection. *OXSM* has been studied in the mouse kidney, and its function is related to glucose reabsorption [60]. Biological reabsorption in the kidney is important because it enables the full use of nutrients and also helps to regulate the pH of fluids in the body. NGLY1 is related to cerebral visual impairment (CVI) [61] in humans. CVI is a collective term that includes several visual disorders that result from damage to, or malfunction of, cerebral components of the visual system, such as the optic tracts, optic radiations, and the visual cortex [59]. The functions of these two genes suggest that they may have been targets for artificial selection, that is, introgressed genes had been retained by artificial selection.

When D-values are used to find introgressed regions, outlier values occur as statistical noise [36], generated mainly in regions of low diversity. Therefore, in this experiment, we used D-values only as evidence for introgression in Chinese local breeds. More generally for introgressed regions, fD values were used to correct for biases introduced by outlier values. Bias was also reduced by using multiple populations to calculate fD values and by focusing on the intersection of the introgressed regions in multiple populations. Finally, the results were analyzed using selective scanning to minimize the effects of outliers as much as possible. Although the D-statistic had been used in many studies, there are other methods available when calculating gene flows, such as Hidden Markov Models [62], conditional random fields [63], S*-statistic [64], and similar implementations [65].

Our study revealed introgression from Western commercial pigs to indigenous Chinese breeds using genome-wide markers. Re-sequencing data offers an opportunity to explore introgression in indigenous Chinese breeds in more detail in the future.

## Figures and Tables

**Figure 1 genes-11-00275-f001:**
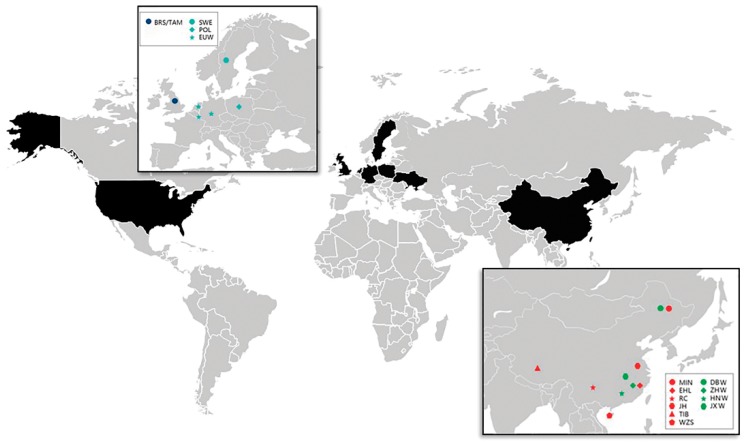
Origins of the samples used in this study. Red icons represent Chinese domesticated pigs, green icons represent Chinese wild boars, blue icons represent European domesticated pigs, and cyan icons represent European wild boars. Commercial breeds are not shown.

**Figure 2 genes-11-00275-f002:**
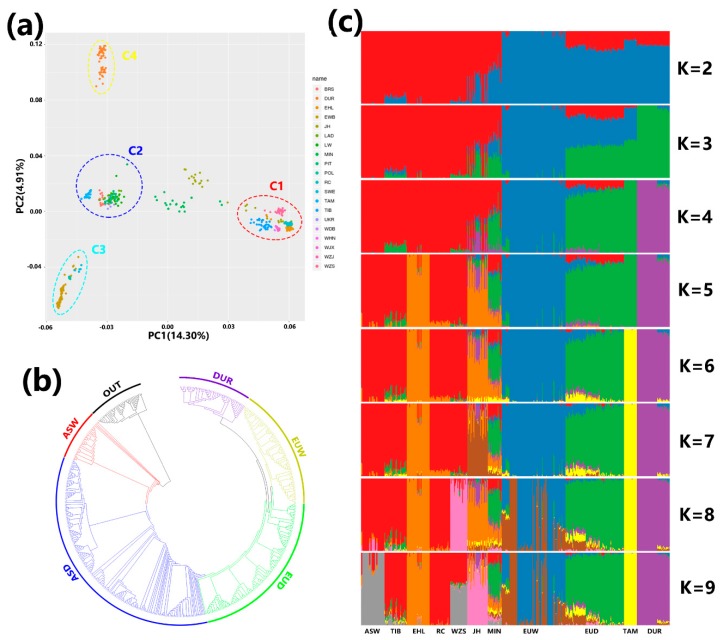
(**a**) Principal component analysis of 486 individuals from the 19 breeds under study. C1: Chinese breeds; C2: European domesticated pigs; C3: European wild boars; C4: Duroc. (**b**) Neighbor-joining tree showing relationships among the 20 populations. (**c**) Bar plot showing ancestry composition generated using ADMIXTURE with the assumed number of ancestries (K) varying from 2 to 9. ASW (Asian wild boar) includes DBW (Dongbei boara), HNW (South China boar), JXW (Jiangxi boar), and ZJW (Zhejiang boar). EUW (European wild boar) includes POL (Polish boar) and EUW (other European boar). EUD (European domesticated pig) includes BRS (British Saddleback), LAD (Landrace), LW (Large White), and PIT (Pietrain).

**Figure 3 genes-11-00275-f003:**
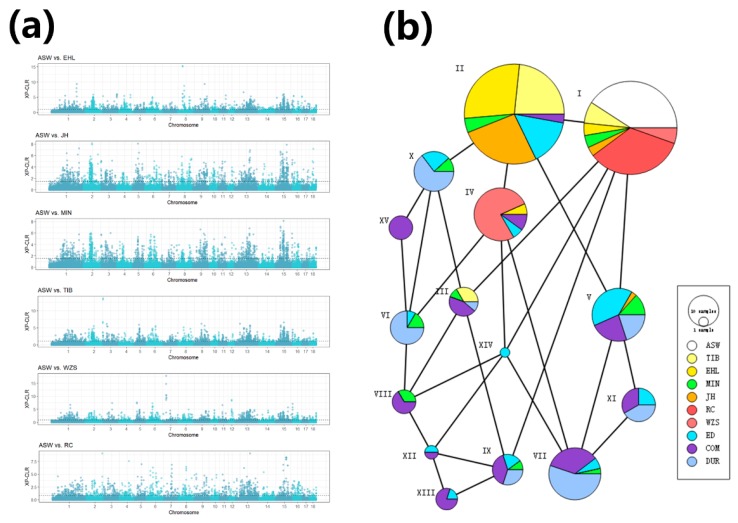
(**a**) XP-CLR (cross-population composite likelihood ratio) values calculated between each indigenous Chinese breed vs. Chinese wild boars. The black dashed lines represent the 5% cutoff used to define selective signals. (**b**) Haplotype network generated using 15 common haplotypes in chr13: 12.5–13.1Mb. Circle area is proportional to the number of samples, and lines between the circles represent a step mutation.

**Table 1 genes-11-00275-t001:** Pig populations and samples used in this study.

Breed	Code	Origin	Number *
Dongbei boar	DBW	Northeast China	3
Zhejiang boar	ZJW	Zhejiang province, China	15
South China boar	HNW	South of China	15
Jiangxi boar	JXW	Jiangxi province, China	5
Tibetan pig	TIB	Tibet A.R., China	35
Erhualian pig	EHL	Jiangsu province, China	36
Jinhua pig	JH	Zhejiang province, China	44
Min pig	MIN	Northeast China	22
Rongchang pig	RC	Sichuan province, China	32
Wuzhishan pig	WZS	Hainan province, China	27
Polish boar	PLW	Poland	6
Swedish boar	SWE	Sweden	6
British Saddleback ^#^	BRS	United Kingdom	20
Tamworth ^#^	TAM	United Kingdom	20
Landrace ^#^	LAD	Europe	20
Large White ^#^	LW	United Kingdom	20
Pietrain ^#^	PIT	Belgium	20
Other European boar ^#^	EUW	Europe	88
Duroc ^#^	DUR	United States	52
Warthogs ^+^	OUT	Africa	39
**Total**			**525**

* Number of samples, # data obtained from public database (http://dx.doi.org/10.5061/dryad.v6f1g), + data obtained from public database (http://dx.doi.org/10.5061/dryad.30tk6).

**Table 2 genes-11-00275-t002:** D-statistics for six indigenous Chinese breeds vs. four Western populations.

	COM	*p*-Value	ED	*p*-Value	DUR	*p*-Value	EW	*p*-Value
EHL	0.03	2.2 × 10^−03^	0.03	1.1 × 10^−03^	0.03	9.0 × 10^−04^	0.03	0.02
TIB	0.12	2.2 × 10^−16^	0.13	6.1 × 10^−15^	0.11	2.2 × 10^−16^	0.61	1.0 × 10^−26^
WZS	0.03	4.9 × 10^−04^	0.02	0.01	0.03	5.9 × 10^−04^	0.04	5.4 × 10^−04^
RC	1.9 × 10^−03^	0.43	5.3 × 10^−03^	0.30	1.4 × 10^−05^	0.49	3.8 × 10^−03^	0.07
JH	0.24	1.0 × 10^−26^	0.26	1.0 × 10^−26^	0.29	1.0 × 10^−26^	0.40	1.0 × 10^−26^
MIN	0.45	1.0 × 10^−26^	0.43	1.0 × 10^−26^	0.42	1.0 × 10^−26^	0.61	1.0 × 10^−26^

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
