# Peer review of "Genome-Wide Analysis Reveals Human-Mediated Introgression from Western Pigs to Indigenous Chinese Breeds"

_genes, 2020, doi:10.3390/genes11030275_

Round 1

Reviewer 1 Report

In this paper, genotype data from a multiple populations of pigs from European and Chinese breeds are analyzed with a focus on introgression and selection. In principle, the study design is good, but I do have a few points which in my opinion need to be addressed.

First and foremost, I am unsure about the power of the sliding window D-statistics or fd-statistics, for actually two independent reasons.

1) First, the publication referred to (Martin et al., 2015) does emphasize that, even though fd is better in inferring introgressed segments, outliers are subject to biases. In this case, I do not propose new analyses or different methods, but this limitation should be mentioned at least in the discussion. There are better methods for detecting introgressed segments, compare for example as a recent review Fontsere et al., 2019.

2) Second, and more concerning, it seems to me that the data density might not be very high. Given a genome size of 2.7 Gbp and ~43k SNPs, the average 100 kbp window would contain only 1.6 SNPs. Of course, SNPs are unequally distributed, but this seems very low. The authors should provide statistics on how many windows don’t have sufficient data, a histogram of sites per window (one could think of a threshold for that as well), or other evidence that the obtained statistics are meaningful given the data density. Also, it is not explained how the Z-transformed fd value was used for p-value estimation.

3) Following that, is it justified to use 100 kbp windows? With ~180 years and a generation time of 5 years, less than 40 generations have passed since the introgression event. Then, larger genomic regions would be introgressed, and it might be better to increase the window size if data density is low. Alternatively, one could think of windows not defined by physical distance, but by number of SNPs, i.e. fd-statistics for 50 consecutive SNPs or something like that. This could even be determined by the average expected length of introgressed haplotypes after 35 generations.

4) What is the rationale of using this type of D-statistic: Asian wild, Asian domestic; European domestic, warthog? In my opinion, this would also detect gene flow from Asian domestic into European domestic groups (allele sharing of the two groups), which has been shown before. Since the European wild boar is the source population for domestication, it would be absolutely necessary to confirm the D-statistics using a setting of Asian wild, Asian domestic; European wild, warthog. From the historical records discussed in the paper, it seems clear that the introgression happened, so I don’t doubt the general results, but the presented statistics will conflate the bidirectional gene flow. For the main result, a robust setting should be provided.

5) Concerning the selection scan, I see the same issue as with the windowed statistics: The window size seems small for the amount of SNPs. Is there any window with more than 200 SNPs, mentioned in the methods section?

6) I highly recommend to create haplotype networks for the main candidate regions (compare prominently Fig. 3 for Tibetans in Huerta-Sanchez et al., 2014). This would strongly visually support the findings.

Smaller points:

7) line 168: what is an “efficient D statistic”?

8) line 239: Neanderthal introgression occurred to all non-African populations, not only European people; a more recent reference would be Prüfer et al., 2017, suggesting 1.8-2.4% in European and 2.3-2.6% in East Asian populations.

Beyond the scientific content, two points:

9) I see no data availability statement. For a fully open access publication, the data should be publicly available as well. Given the size of the dataset, it might be possible as supplementary file, otherwise it should be another database, or FigShare.

10) I think that the text should be corrected by someone with high professional proficiency in English. It is understandable, but needs some editing throughout.

Author Response

Response to Reviewer 1 Comments

Point 1: First, the publication referred to (Martin et al., 2015) does emphasize that, even though fd is better in inferring introgressed segments, outliers are subject to biases. In this case, I do not propose new analyses or different methods, but this limitation should be mentioned at least in the discussion. There are better methods for detecting introgressed segments, compare for example as a recent review Fontsere et al., 2019.

Response 1: We now describe the limitations of the fd-statistic in the Discussion section. See lines 268-276.

Point 2: Second, and more concerning, it seems to me that the data density might not be very high. Given a genome size of 2.7 Gbp and ~43k SNPs, the average 100 kbp window would contain only 1.6 SNPs. Of course, SNPs are unequally distributed, but this seems very low. The authors should provide statistics on how many windows don’t have sufficient data, a histogram of sites per window (one could think of a threshold for that as well), or other evidence that the obtained statistics are meaningful given the data density. Also, it is not explained how the Z-transformed fd value was used for p-value estimation.

Response 2: Thank you for your suggestion, now the windows size was modified for encompassing a constant number of SNPs, and 10 SNPs are included in the window (see detailed answer to Point 3).

To compare all populations uniformly, we performed a Z-transform on the results, that is, we transformed the results into a standard normal distribution. The formula for the Z-transform is , where  is the modified fD-statistic,  is expected value of  for five introgressed populations, and  is standard deviation. This transform has been used in other similar studies (Teng et.al. 2017, Hu et.al. 2018). We now describe this analysis in more detail in the revised version of the manuscript. See lines 113-115.

  • Teng, H.; Zhang, Y.; Shi, C.; Mao, F.; Cai, W.; Lu, L.; Zhao, F.; Sun, Z.; Zhang, J. Population genomics reveals speciation and introgression between brown Norway rats and their sibling species. Molecular biology and evolution 2017, 34, 2214–2228.
  • Hu, X.-J.; Yang, J.; Xie, X.-L.; Lv, F.-H.; Cao, Y.-H.; Li, W.-R.; Liu, M.-J.; Wang, Y.-T.; Li, J.-Q.; Liu, Y.-G.; et al. The genome landscape of tibetan sheep reveals adaptive introgression from argali and the history of early human settlements on the Qinghai–Tibetan plateau. Molecular biology and evolution 2018, 36, 283–303.

Point 3: Following that, is it justified to use 100 kbp windows? With ~180 years and a generation time of 5 years, less than 40 generations have passed since the introgression event. Then, larger genomic regions would be introgressed, and it might be better to increase the window size if data density is low. Alternatively, one could think of windows not defined by physical distance, but by number of SNPs, i.e. fd-statistics for 50 consecutive SNPs or something like that. This could even be determined by the average expected length of introgressed haplotypes after 35 generations.

Response 3: Thank you for your suggestion concerning window size. Windows are now defined to contain a constant number of SNPs and three different sliding window sizes(10 SNPs, 20 SNPs and 30 SNPs per windows) were tested After comparing results of different window sizes, finally a window containing 10 consecutive SNPs (with a step size of 2 SNPs) across the genomes was applied in this study. This yields an average physical length for the fd windows of 1.09 Mb. The windowing strategy is described at lines 109-111 and 176-178.

  • Ai, H.; Huang, L.; Ren, J. Genetic diversity, linkage disequilibrium and selection signatures in Chinese and Western pigs revealed by genome-wide SNP markers. PloS one 2013, 8, e56001.
  • Cahill, J.A.; Stirling, I.; Kistler, L.; Salamzade, R.; Ersmark, E.; Fulton, T.L.; Stiller, M.; Green, R.E.; Shapiro, B. Genomic evidence of geographically widespread effect of gene flow from polar bears into brown bears. Molecular ecology 2015, 24, 1205–1217.

Point 4: What is the rationale of using this type of D-statistic: Asian wild, Asian domestic; European domestic, warthog? In my opinion, this would also detect gene flow from Asian domestic into European domestic groups (allele sharing of the two groups), which has been shown before. Since the European wild boar is the source population for domestication, it would be absolutely necessary to confirm the D-statistics using a setting of Asian wild, Asian domestic; European wild, warthog. From the historical records discussed in the paper, it seems clear that the introgression happened, so I don’t doubt the general results, but the presented statistics will conflate the bidirectional gene flow. For the main result, a robust setting should be provided.

Response 4: We thank the reviewer for raising this point. We confirmed the D-statistics using a setting of Asian wild, Asian domestic; European wild, warthog  according to reviewer’s comment, and found  that the results were consistent with previous one, five introgressed populations still have a significant p-values, which  suggests that the introgression are from Western pigs. We modified this part in the revised manuscript at lines 170-173.

Point 5: Concerning the selection scan, I see the same issue as with the windowed statistics: The window size seems small for the amount of SNPs. Is there any window with more than 200 SNPs, mentioned in the methods section?

Response 5: As described above (see Point 3), the fd-statistic window has been redefined to contain a constant number of consecutive SNPs. We have also modified the windows size in the selection scan. The window size has been expanded to 1 Mb with a 200 kb step size. This is described in the revised manuscript at lines 122-123 and 191-193.

And the window with more than 200 SNPs, which mentioned in the method section was a mistake, which was corrected in the new version, thanks for pointing out.

Point 6: I highly recommend to create haplotype networks for the main candidate regions (compare prominently Fig. 3 for Tibetans in Huerta-Sanchez et al., 2014). This would strongly visually support the findings.

Response 6: Thank you for this suggestion. We have incorporated a haplotype network into the Results section (see lines 204-210 and Figure 3b).

Point 7: line 168: what is an “efficient D statistic”?

Response 7: The phrase “efficient D statistic” has been corrected to “D statistic” at line 168.

Point 8: line 239: Neanderthal introgression occurred to all non-African populations, not only European people; a more recent reference would be Prüfer et al., 2017, suggesting 1.8-2.4% in European and 2.3-2.6% in East Asian populations.

Response: Thank you. We have updated the corresponding text at lines 246-248.

Point 9: I see no data availability statement. For a fully open access publication, the data should be publicly available as well. Given the size of the dataset, it might be possible as supplementary file, otherwise it should be another database, or FigShare.

Response 9: We now provide our data in the supplementary material.

Point 10: I think that the text should be corrected by someone with high professional proficiency in English. It is understandable, but needs some editing throughout.

Response 10: The revised manuscript has been checked and corrected for English grammar and overall readability by a professional editing service.

Reviewer 2 Report

This could be a potentially interesting paper since introgression of European breeds into Asia is far less well characterized than Chinese introgression into Europe. The analysis is, however, somewhat limited and is poorly written. Please put data in perspective also as this has also been analyzed previously (Yang et al, GSE). Some analysis are perhaps not needed, eg, the Fst analysis are somewhat redundant to PCA. A local Fst would have been much more useful.

Introgression is a widespread phenomenon across numerous domestic species, in pigs, this is also studied in a different context in American creole breeds (https://www.nature.com/articles/hdy2012109). 

Specify both in abstract and introduction that the genome data refer to SNP array data.

line 71: Specify which samples are new and which are public. Also in table 1. This part is absolutely not clear. Are new SNP data available?

Treemix is highly unreliable and provides no real insight, please remove. Explain briefly basis of XP-CLR, and how was it applied, what breeds mere compared against? How sensitive is it to admixture? 

Why were SNPs failing HWE removed? You do not expect HWE in structured populations.

Line 38: add reference on Asia domestication

l133: This is widely known already.

l142: why as expected?

l178: How many SNPs are there per window, on average?

l193: all China pigs in a pool? Were Chinese wild boar from both North and South?

l200: This is potentially interesting, but why not simply a shared ancestry due to a bottleneck rather than selection?

How significance is computed in XP-CLR test? 

Paper MUST be rewritten with proper English.

Author Response

Response to Reviewer 2 Comments

Point 1: This could be a potentially interesting paper since introgression of European breeds into Asia is far less well characterized than Chinese introgression into Europe. The analysis is, however, somewhat limited and is poorly written. Please put data in perspective also as this has also been analyzed previously (Yang et al, GSE). Some analysis are perhaps not needed, eg, the Fst analysis are somewhat redundant to PCA. A local Fst would have been much more useful.

Introgression is a widespread phenomenon across numerous domestic species, in pigs, this is also studied in a different context in American creole breeds (https://www.nature.com/articles/hdy2012109).

Response 1: We agree that the Fst analysis is not required and have removed it from the revised manuscript. For the literature you mentioned, we have updated the corresponding text at line 50 and line 54.

Point 2: Specify both in abstract and introduction that the genome data refer to SNP array data.

Response 2: We have modified the manuscript to make it clear in the abstract and introduction that a SNP array is being used to generate our genome data.

Point 3: line 71: Specify which samples are new and which are public. Also in table 1. This part is absolutely not clear. Are new SNP data available?

Response 3: Thank you for this suggestion. We now carefully differentiate between the 278 samples that are new (i.e., generated by us), and other data that were obtained from public sources. The new SNP data have been included in the supplementary materials.

Point 4: Treemix is highly unreliable and provides no real insight, please remove. Explain briefly basis of XP-CLR, and how was it applied, what breeds mere compared against? How sensitive is it to admixture?

Response 4: We agree with your reservations about Treemix, and have removed the Treemix analysis from the revised manuscript. Concerning XP-CLR, this is a multi-locus sliding window test that jointly models the multilocus allele frequency differentiation between two populations (Chen et.al. 2010). This method has been applied in studies of population genetics for humans (Scheinfeldt et.al. 2012, Udpa et.al. 2014), the house mouse (Staubach et.al. 2012), and maize (Hufford et.al. 2012). In our study, the XP-CLR was used to compare the differentiation between each introgressed indigenous Chinese breed vs. the Chinese wild boar. The analysis is expected to reveal regions affected by artificial selection. The XP-CLR statistic is particularly robust to ascertainment bias and population demography, and we used it to determine whether introgressed regions had also been affected by artificial selection. We have expanded our description of XP-CLR in the revised manuscript in lines 117-120 and 191-193.

  • Chen, H.; Patterson, N.; Reich, D. Population differentiation as a test for selective sweeps. Genome research 2010, 20, 393–402.
  • Scheinfeldt, L.B.; Soi, S.; Thompson, S.; Ranciaro, A.; Woldemeskel, D.; Beggs, W.; Lambert, C.; Jarvis, J.P.; Abate, D.; Belay, G.; et al. Genetic adaptation to high altitude in the Ethiopian highlands. Genome biology 2012, 13, R1.
  • Udpa, N.; Ronen, R.; Zhou, D.; Liang, J.; Stobdan, T.; Appenzeller, O.; Yin, Y.; Du, Y.; Guo, L.; Cao, R.; et al. Whole genome sequencing of Ethiopian highlanders reveals conserved hypoxia tolerance genes. Genome biology 2014, 15, R36.
  • Staubach, F.; Lorenc, A.; Messer, P.W.; Tang, K.; Petrov, D.A.; Tautz, D. Genome patterns of selection and introgression of haplotypes in natural populations of the house mouse (Mus musculus). PLoS Genetics 2012, 8.
  • Hufford, M.B.; Xu, X.; Van Heerwaarden, J.; Pyhäjärvi, T.; Chia, J.-M.; Cartwright, R.A.; Elshire, R.J.; Glaubitz, J.C.; Guill, K.E.; Kaeppler, S.M.; et al. Comparative population genomics of maize domestication and improvement. Nature genetics 2012, 44, 808.

Point 5: Why were SNPs failing HWE removed? You do not expect HWE in structured populations.

Response 5: Thank you for your comment. After carefully consideration, we agree with your idea and removed the conditions of HWE. Now 42819 SNPs remain after quality control. Related contents have been modified in line 75-78, please read our new version.

Point 6: Line 38: add reference on Asia domestication

Response 6: Thank you for catching this missing reference. The citation has been added to the revised manuscript line 37.

Point 7: l133: This is widely known already.

Response 7: The corresponding text has been modified in the revised manuscript.

Point 8: l142: why as expected?

Response 8 Thanks. We obtained four clusters from the PCA results, and therefore we expect that similar results would be generated in a gene structure analysis for small K values (K<=4). We have modified the manuscript to make this point more clearly in 146-148.

Point 9: l178: How many SNPs are there per window, on average?

Response 9: We modified the window size in the new version, now the window size has been expanded to 1 Mb with a step size of 200 kb step. These windows contain 13.68 SNPs on average.

Point 10: l193: all China pigs in a pool? Were Chinese wild boar from both North and South?

Response 10: We calculated XP-CLR values between the five introgressed populations (MIN, JH, WZS, EHL, and TIB) and Chinese wild boars. The wild boars sampled were from the north and south of China, specifically Northeast China, Jiangxi Province, and Zhejiang Province (Figure 1). This information has been added to the revised manuscript in lines 70-71 for better understanding.

Point 11: l200: This is potentially interesting, but why not simply a shared ancestry due to a bottleneck rather than selection?

Response 11: Our goal was to find regions that have been subjected both to introgression and artificial selection. In these regions, the introgressed genotypes have been retained even after artificial selection. We agree that a bottleneck potentially explains the observations. However, our conclusion is based on 5 independent XP-CLR tests between indigenous pigs (5 breeds, all of which have been subjected to introgression) and Chinese wild boars.  All five breeds differed from Chinese wild boars in the introgressed regions, and those five breeds originated in different geographic areas (Table 1). Therefore, we think this is more likely due to artificial selection.

Point 12: How significance is computed in XP-CLR test?

Response 12: We did not compute significance using a statistical test, but instead focused attention on regions with the highest (top 5%) XP-CLR values. These regions are substantially different between breeds. By calculating the common selection regions that had also been subjected to introgression in the 5 breeds, we obtained 19 artificial selection regions. These results are now described in more detail in revised manuscript lines 125-126 and 193-194.

Point 13: Paper MUST be rewritten with proper English.

Response 13:The revised manuscript has been checked and corrected for English grammar and overall readability by a professional editing service.

Round 2

Reviewer 1 Report

Everything has been addressed appropriately, the manuscript has greatly improved.

Author Response

Point1: Everything has been addressed appropriately, the manuscript has greatly improved.

Response 1: Thanks for your kind help.

Reviewer 2 Report

I have no major concern and I believe the paper can be of interest, especially if new SNP data are made public.

MINOR

l23: had been introgressed

l25: breeds SHOW EVIDENCE OF introgression

l85 show public website

Remove unnecessary digits in Table and text, eg, P-value = 0.43448 --> 0.43

l191 genes POSSIBLY under

l193: kb k is lower case

l218 higher levels of inrogression , also shown by Yang et al GSE

l278-280: remove 'A large ... may be biased'

Author Response

Point 1: l23: had been introgressed

Response 1: The text has been modified in the revised manuscript line 23.

Point 2: l25: breeds SHOW EVIDENCE OF introgression

Response 2: The text has been modified in the revised manuscript lines 25-26.

Point3: l85 show public website

Response 3: The public website has been added in lines 85-86.

Point 4: Remove unnecessary digits in Table and text, eg, P-value = 0.43448 --> 0.43

Response 4: The digits have been removed in lines 168, 175, 176 and Table 2.

Point 5: l191 genes POSSIBLY under

Response 5: The text has been modified in the revised manuscript line 192.

Point 6: l193: kb k is lower case

Response 6: We has modified to kb in the revised manuscript line 194.

Point 7: l218 higher levels of inrogression , also shown by Yang et al GSE

Response 7: Thank you for the citation, we have updated the corresponding text at line 219.

Point 8: l278-280: remove 'A large ... may be biased'

Response 8: We have updated the corresponding text at lines 278-280.